# Association of ABO blood groups with presentation and outcomes of confirmed SARS CoV-2 infection: A prospective study in the largest COVID-19 dedicated hospital in Bangladesh

**Reaz Mahmud**[1][☮][*], **Mohammad Aftab Rassel**[1‡], **Farhana Binte Monayem**[2‡], **S. K. Jakaria Been Sayeed**[3☮¤a], **Md Shahidul Islam**[2‡], **Mohammed Monirul Islam**[4‡], **Mohammad Abdullah Yusuf**[5‡], **Sabrina Rahman**[3‡], **K. M. Nazmul Islam**[6‡], **Imran Mahmud**[3‡], **Mohammad Zaid Hossain**[3‡], **Ahmed Hossain Chowdhury**[1‡], **A. K. M. Humayon Kabir**[3‡], **Kazi Gias Uddin Ahmed**[1‡], **Md. Mujibur Rahman**[3☮¤b]

1 Department of Neurology, Dhaka Medical College, Dhaka, Bangladesh, 2 Sarkari karmachari Hospital, Dhaka, Bangladesh, 3 Department of Medicine, Dhaka Medical College, Dhaka, Bangladesh, 4 Ministry of Health and Family Planning Welfare, Dhaka, Bangladesh, 5 Department of Microbiology, National Institute of Neurosciences and Hospital, Dhaka, Bangladesh, 6 Department of Neurology, Shaheed Suhrawardy Medical College, Dhaka, Bangladesh

☮ These authors contributed equally to this work.
¤a Current address: Stroke Unit, National Institute of Neurosciences and Hospital, Dhaka, Bangladesh
¤b Current address: Department of Medicine, Bangabandhu Sheikh Mujib Medical University, Shahbag, Bangladesh
‡ These authors also contributed equally to this work
* reazdmc22@yahoo.com

**Data Availability Statement:** All relevant data are within the paper and its Supporting Information

## Abstract

### Background

Globally, studies have shown conflicting results regarding the association of blood groups with SARS CoV-2 infection.

### Objective

To observe the association between ABO blood groups and the presentation and outcomes of confirmed COVID-19 cases.

### Design, setting, and participants

This was a prospective cohort study of patients with mild-to-moderately severe COVID-19 infections who presented in the COVID-19 unit of Dhaka Medical College Hospital and were enrolled between 01 June and 25 August, 2020. Patients were followed up for at least 30 days after disease onset. We grouped participants with A-positive and A-negative blood groups into group I and participants with other blood groups into group II.

files as well as deposited to Dryad data repository (DOI: https://doi.org/10.5061/dryad.dv41ns1xk).

**Funding:** The Authors received no specific Funding for this work.

**Competing interests:** The authors have declared that no competing interests exist.

## Results

The cohort included 438 patients; 52 patients were lost to follow-up, five died, and 381 completed the study. The prevalence of blood group A [144 (32.9%)] was significantly higher among COVID-19 patients than in the general population (p < 0.001). The presenting age [mean (SD)] of group I [42.1 (14.5)] was higher than that of group II [38.8 (12.4), p = 0.014]. Sex (p = 0.23) and co-morbidity (hypertension, p = 0.34; diabetes, p = 0.13) did not differ between the patients in groups I and II. No differences were observed regarding important presenting symptoms, including fever (p = 0.72), cough (p = 0.69), and respiratory distress (p = 0.09). There was no significant difference in the median duration of symptoms in the two group (12 days), and conversion to the next level of severity was observed in 26 (20.6%) and 36 patients (13.8%) in group I and II, respectively. However, persistent positivity of RT-PCR at 14 days of initial positivity was more frequent among the patients in group I [24 (19%)] than among those in group II [29 (11.1%)].

## Conclusions

The prevalence of blood group A was higher among COVID-19 patients. Although ABO blood groups were not associated with the presentation or recovery period of COVID-19, patients with blood group A had delayed seroconversion.

## Introduction

The World Health Organization (WHO) first reported the emergence of COVID-19 infection in Wuhan City, China in late December [1]. Subsequently, the disease caused by this novel virus was declared to be a pandemic on March 12, 2020 [2]. As of September 19, 2020, there have been 30,369,778 confirmed cases of COVID-19, including 948,795 deaths reported to the WHO [3]. The presentation of COVID-19 varies widely. The most common symptoms of COVID-19 include fatigue, fever, dry cough, respiratory distress, and anosmia. Moreover, a large proportion of patients remain asymptomatic [4]. Its behavior also shows regional variation. A genome-wide association study on severe COVID-19 with respiratory failure detected cross-replicating associations with rs11385942 at locus 3p21.31 and with rs657152 at locus 9q34.2. The association signal at locus 9q34.2 coincided with the ABO blood group locus. They found a higher risk in individuals with blood group A than in those with other blood groups [5]. Other viruses, such as HBV [6], SARS-COV [7], and MERS-COV [8], are susceptible to ABO blood groups. The mechanisms underlying the association between the blood groups are still unclear. Histo-blood group antigens are expressed on endothelial cells (ECs) and platelets [9]. The SARS-CoV-2 enters the human body through replication in epithelial cells of the respiratory and digestive tracts. They have the ability to synthesize ABH carbohydrate epitopes. It has been hypothesized that the S protein of the virions produced by either A or B individuals can be decorated with A or B carbohydrate epitopes, respectively. Guillon et al. [10] reported that the interaction between the S protein and angiotensin-converting enzyme 2 (ACE-2) is specifically inhibited by human natural anti-A antibodies. SARS-CoV and SARS-CoV-2 have similar nucleic acid sequences and similar receptor combinations with ACE-2 [11]. It was also found that the non-A type is a risk factor for venous thromboembolism, which is one of the causes of COVID-19 deaths [12]. However, different studies have shown contradictory results regarding the influence of blood groups on the susceptibility and

outcome of COVID-19 infections [13–16]. Therefore, this study was conducted to observe the association of ABO blood groups with the presentation and outcome of confirmed COVID-19 infections in the largest COVID-19 dedicated hospital in Bangladesh.

## Materials and methods

This single-center prospective cohort study was conducted to evaluate the association of the ABO blood groups with the presentation and outcomes of confirmed COVID -19 infections in hospitalized and outdoor patients with COVID-19. The outcomes in this research included: A. Duration required for clinical improvement, as defined below; B. Proportion of patients converted to the next level of severity; C. Proportion of the patients remaining positive for RT-PCR of COVID-19 on day 14 after the initial positivity; D. Development of post-COVID syndrome as defined below. The participants were enrolled from 01 June to 25 August, 2020. The study was conducted in the Department of Medicine, Dhaka Medical College Hospital. The recruitment was limited to patients who were more than 18 years of age with confirmed COVID-19 (RT-PCR positive) infection. Patients with hemoglobinopathies or other blood disorders were excluded from the study. Written informed consent was obtained from all the patients. Ethical approval was obtained from the ethical review committee of the institute. The capacity to provide consent was determined by the investigators in the presence of an attendant, a testimony of the attendant was obtained in the consent form, and the ethical review committee approved this consent procedure during the approval of the protocol. Those who were minors or unable to provide consent were excluded from the study. For the estimation, we grouped A positive and A negative blood groups into group I and other blood groups, such as B, AB, and O, irrespective of their Rh status, into group II. An assumption that the expected proportions to be cured from COVID-19 by day 12 in group I (blood group A) and group II (blood groups B, O, AB) are 0.70 and 0.90, respectively. Thus, we required a total of 378 samples at a 1:2 ratio, which would provide a power of at least 90% in two-tailed tests and a p value less than 0.05, to detect significant differences between the groups. Therefore, considering a 10% dropout rate, we needed 416 samples in total.

### Procedure

A case record form was constructed to collect baseline information, such as demographics, blood groups, clinical signs and symptoms, comorbidities, and oxygen saturation. Routine investigations, including CBC, ESR, CRP, creatinine, RBS, SGPT, chest X-ray, and D-dimer were advised on enrollment. Real-time-polymerase chain reaction testing for COVID-19 was performed 14 days after the initial positive test in all patients. The patients were followed up directly or over the telephone with at least three-day intervals up to 30 days from the onset of the disease. Clinical improvement in patients was assessed according to the improvement criteria of the WHO and Bangladesh guidelines [17,18], which required that the body temperature remained normal for at least 3 days, respiratory symptoms were significantly improved (respiratory rate < 25 and no dyspnea), and $SpO_2$ >93% was achieved without assisted oxygen inhalation. Mild disease was defined as the symptoms of an upper respiratory tract viral infection, including mild fever, cough (dry), sore throat, nasal congestion, malaise, headache, muscle pain, anosmia, or malaise. Moderate disease, including respiratory symptoms, such as cough and shortness of breath are present without signs of severe pneumonia. Severe disease included severe dyspnea, tachypnea (> 30 breaths/min), and hypoxia ($SpO_2$ < 90% in room air). These classifications were made according to the World Health Organization and national guidelines of Bangladesh [17,18]. In this study, we assessed the proportion of patients with early recovery (clinical improvement within 7 days of symptom onset), late recovery (clinical

improvement required ≥12 days), severity conversion (patients progress to more serious disease), persistently positive for RT-PCR of COVID-19 (positive RT-PCR on a 14 day test), and post-COVID syndrome (in the absence of any definition, we defined it as 1. Persistence of illness with signs and symptoms beyond virologic clearance 2. Development of new symptoms within 1 month after the initial clinical and virologic cure, the etiology of which is postulated to be viral infection.

## Statistical analysis

The data were analyzed using STATA version-15.1, StataCorp, 4905 Lakeway Drive, College Station, Texas 77845 USA. To compare the variables between group I (blood group A) and group II (other blood groups), an independent sample t-test was used. The chi-square test was used to analyze the categorical variables. For the outcome assessment, relative risk (RR) with 95% CI was measured for qualitative variables categorizing groups I and II. The difference in the median time-to-recovery between group I and group II was determined with cumulative incidence by blood group from the Fine-Gray Model. The hazard ratios were calculated using cause-specific hazard models. Categorical variables are presented as n (%), normally distributed continuously as mean (SD), and skewed continuous variables as median (IQR). The statistical significance was set at P <0.05.

## Results

Of the 554 patients who were screened and assessed for eligibility, a total of 438 patients were enrolled in the study. A total of 52 patients were lost to follow-up, 5 patients died, and 381 patients completed the follow-up. The analysis was performed on 438 patients (Fig 1).

Among them, 188 patients were admitted to the hospital. The mean age of the patients was 39.8 (13). The majority of the participants were younger than 40 years [266 participants (60.6%)] and men were predominantly affected [258 (58.9%)]. Most of the patients presented with fever (333 patients, 76%) and cough (274 patients, 62.6%). Respiratory distress developed

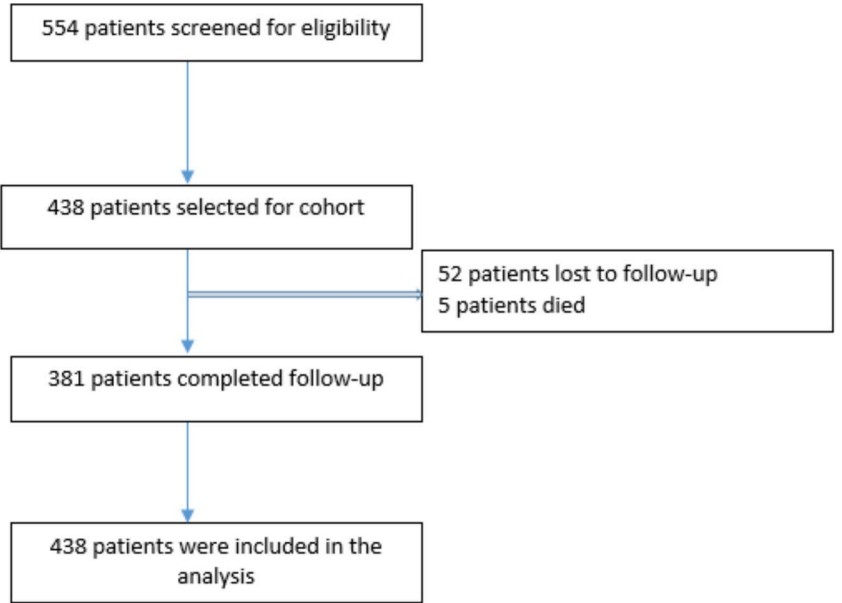

**Fig 1. Sample selection in this prospective cohort study in adults with COVID-19 infection.**

**Table 1. Baseline characteristics of COVID-19 patients with different blood groups (N = 438).**

| Variables | Total (N = 438) | Group I[c] (N = 144) | Group II[d] (N = 294) | p-value |
|---|---|---|---|---|
| Age (years) Mean (SD) | 39.8(13.2) | 42.1(14.5) | 38.8(12.4) | 0.014 |
| Sex (Male), n (%) | 258(58.9) | 79(54.9) | 179(60.9) | 0.23 |
| **Symptoms** | | | | |
| Fever, n (%) | 333(76.0) | 111(77.1) | 222(75.5) | 0.72 |
| Cough, n (%) | 274(62.6) | 92(63.9) | 182(61.9) | 0.69 |
| Running Nose, n (%) | 39(8.8) | 13(9.0) | 26(8.8) | 0.95 |
| Respiratory Distress[a], n (%) | 191(43.6) | 71(49.3) | 120(40.8) | 0.09 |
| Sore throat, n (%) | 101(23.1) | 32(22.2) | 69(23.5) | 0.77 |
| Hoarseness of voice, n (%) | 16(3.7) | 2(1.4) | 14(4.8) | 0.08 |
| Diarrhea, n (%) | 37(8.4) | 9(6.2) | 28(9.5) | 0.25 |
| Vomiting, n (%) | 27(6.2) | 8(5.6) | 19(6.5) | 0.71 |
| Anorexia, n (%) | 122(27.9) | 49(34.0) | 73(24.8) | 0.04 |
| Anosmia, n (%) | 155(35.4) | 42(29.2) | 113(38.4) | 0.06 |
| Headache, n (%) | 95(21.7) | 29(20.1) | 66(22.4) | 0.58 |
| Lethargy, n (%) | 112(25.6) | 32(22.2) | 80(27.2) | 0.26 |
| Body ache, n (%) | 73(16.7) | 25(17.4) | 48(16.3) | 0.79 |
| Hypertension, n (%) | 63(14.4) | 24(16.7) | 39(13.3) | 0.34 |
| Diabetes, n (%) | 55(12.6) | 23(16.0) | 32(10.9) | 0.13 |
| **Severity grade[b] at presentation** | | | | |
| Mild, n (%) | 304(69.4) | 94(65.3) | 210(71.4) | 0.19 |
| Moderate, n (%) | 134(30.6) | 50(34.7) | 84(28.6) | |

[a] Shortness of breath, respiratory rate >25/min, or oxygen saturation <93%.

[b] Disease severity at presentation: mild symptoms of an upper respiratory tract viral infection, including mild fever, cough (dry), sore throat, nasal congestion, malaise, headache, muscle pain, anosmia, or malaise.

Moderate respiratory symptoms, such as cough and shortness of breath are present without signs of severe pneumonia (tachypnea > 30 breaths/min, and hypoxia: $SpO_2$ < 90% on room air).

[c] Blood group A positive and negative.

[d] Other blood groups-B, AB, and O including Rh status.

in 191 patients (43.6%) during the disease course. The majority of the patients [304 patients (69.4%)] exhibited mild severity. Overall, 113 (25.7%) patients presented with co-morbidities. Among them, hypertension was present in 63 (14.4%) and diabetes in 55 (12.6%) patients (Table 1).

The average median (IQR) duration of illness was 12 (range, 8–16) days. In Group I, the IQR was 12 days (9–16 days) and Group II it was 12 days (8–15 days) (HR, 95% CI, 1.14, 0.91–1.41, p = 0.25) (Table 2).

The recovery within 7 days was 202 (52.3%) for the total population, 60 (47.6%) in Group II, and 142 (54.6%) in Group II (RR, 1.15; 95% CI, 0.93–1.43; p = 0.20). In 111 (28.8%) patients, the symptoms persisted for more than 12 days. In group I and group II it was 39 (31.0%) and 72(27.7%), respectively (RR, 1.12; 95% CI, 0.81–1.55; p = 0.51). A total of 62 (16.1%) patients progressed to the next level of severity, including 26 patients (20.6%) in group

**Table 2. Duration of illness of COVID-19 patients among Group I and Group II.**

| Attribute | Total (N = 386) | Group I (N = 126) | Group II (N = 260) | Cause-Specific Hazard Ratio 95% CI | p-value |
|---|---|---|---|---|---|
| Total duration of illness, Median (IQR) | 12 (8–16) | 12 (9–16) | 12 (8–15) | 1.14 (0.91–1.41) | 0.25 |

**Table 3. Outcome of COVID-19 patients among Group I and Group II.**

| Attribute | Total (N = 386) n (%) | Group I (N = 126) n (%) | Group II (N = 260) n (%) | Relative Risk 95% CI | p-value |
|---|---|---|---|---|---|
| Recovery[a,b] within 7[§] days | 202 (52.3) | 60 (47.6) | 142 (54.6) | 1.15 (0.93–1.42) | 0.20 |
| Persistence of symptoms 12 days or more | 111 (28.8) | 39 (31.0) | 72 (27.7) | 1.12 (0.81–1.55) | 0.51 |
| Conversion to next level of severity[c] | 62 (16.1) | 26 (20.6) | 36 (13.9) | 1.49 (0.94–2.35) | 0.09 |
| Persistent positivity[d] | 53 (13.7) | 24 (19.0) | 29 (11.1) | 1.71 (1.04–2.81) | 0.04 |
| Post COVID syndrome[e] | 172 (44.6) | 65 (51.6) | 107 (41.1) | 1.25 (1.00–1.57) | 0.05 |

[a] Clinical recovery was defined as a normal body temperature for at least 3 days, improved respiratory symptoms defined as no shortness of breath and respiratory rate <25/min, and an oxygen saturation greater than 93% without supplemental oxygen.

[b] Response criteria was that the patient had recovered clinically as defined above. The day at which clinical recovery started was considered as the response day.

[c] Disease stages were defined as mild symptoms of an upper respiratory tract viral infection, including mild fever, cough (dry), sore throat, nasal congestion, malaise, headache, muscle pain, anosmia, or malaise. Moderate respiratory symptoms, such as cough and shortness of breath are present without signs of severe pneumonia (tachypnea > 30 breaths/min, and hypoxia: $SpO_2 < 90\%$ on room air. Severe tachypnea > 30 breaths/min, and hypoxia: $SpO_2 < 90\%$ on room air.

[d] Persistent RT-PCR positive patient remained positive for RT-PCR at 14 days test.

[e] In the absence of any definition, we defined it as 1. Persistence of illness signs and symptoms beyond virologic clearance 2. New development of symptoms within 1 month after initial clinical and virologic cure, the etiology of which is postulated to be viral infection.

§Days calculated from the onset of symptoms to the day of clinical recovery.

Group I: Blood group A, irrespective of Rh phenotype.

Group II: Blood groups B, AB, and O, irrespective of Rh phenotype.

I and 36 (13.9%) in group II (RR, 1.49; 95% CI, 0.94–2.35; p = 0.09). The number of patients who remained positive even 14 days after initial positivity was 53 (13.7%); among them, 24 patients (19.0%) were in group I and 29 (11.1%) were in group II (RR, 1.71; 95% CI, 1.04–2.81; p = 0.04). After improvement of the initial symptoms, 172 (44.6%) patients developed post-COVID syndrome, including 65 patients (51.6%) in group I and 107 (41.1%) in group II (RR, 1.25; 95% CI, 1.00–1.57; p = 0.05; Table 3).

There were no significant differences in the time to recovery from illness between groups I and II in **terms of the cumulative incidence by blood group from the Fine-Gray Model** (Fig 2).

## Discussion

In the current study, the main aim was to observe the association of the different blood groups on the presenting features and outcomes of mild-to-moderate COVID-19 infections. No differences were observed with respect to the presentation and duration of recovery among the different blood groups. In this study, the majority of the patients were young. It was observed that 60.4% of the affected patients were aged < 40 years, and 8% of the patients were aged > 60 years. Among the affected individuals in this study, 144 patients had blood group A (32.8%), 148 had blood group B (33.7%), 52 had blood group AB (11.9%), and 94 had blood group O (21.5%). In a study [19] conducted by the Blood Bank of Dhaka Medical College, the proportions of different blood groups in the community were as follows: A, 21.8%; B, 37.5%; AB, 8.9%; and O, 31.8%. A higher prevalence of blood group A was observed among the affected individuals. A one-sample t-test revealed a p value of < 0.001. A Turkish study [20] found similar results. Most patients presented with fever, cough, anosmia, anorexia, and lethargy. We did not find any significant differences in symptoms between group I (blood group A) and group II (other blood groups); similar results were found in another study [13]. In this study, comorbidities were present in approximately 26% of the cases. The most important factors were hypertension and diabetes. We observed no statistically significant differences in the presenting features and comorbidities among the patients with different blood groups.

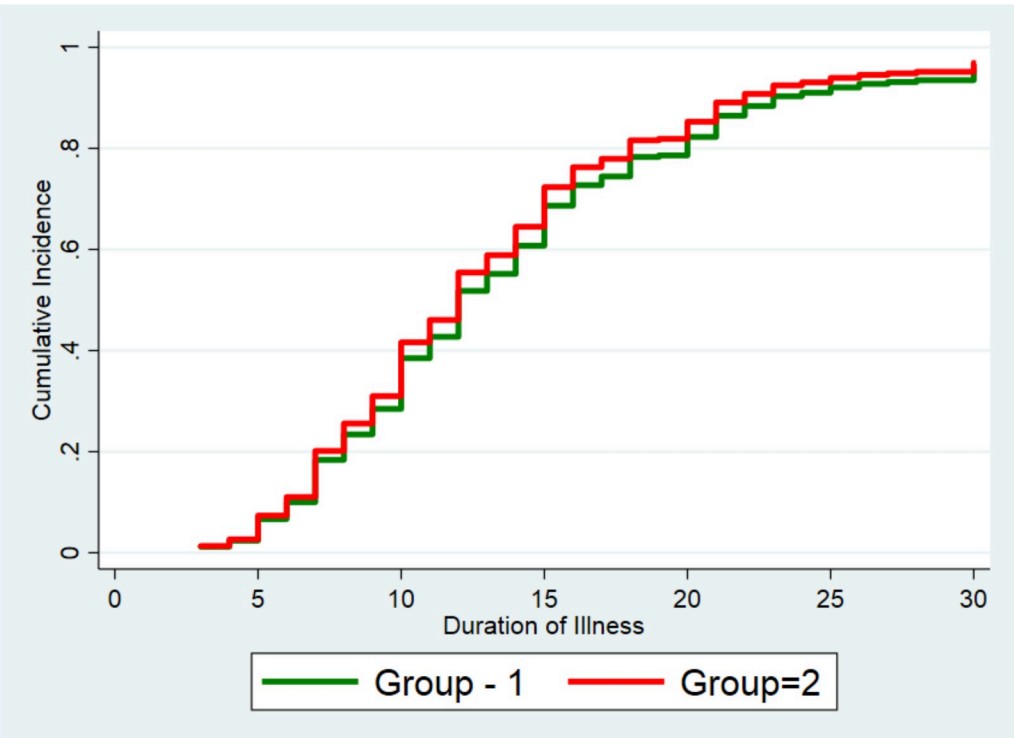

Group I: Blood group A, irrespective of Rh phenotype
Group II: Blood groups B, AB, and O, irrespective of Rh phenotype

**Fig 2. Cumulative incidence by blood group from Fine-Gray Model.**

Globally, approximately 80% of adults present with mild (40%) to moderate illness (40%), 15% with severe illness, and 5% with critical illness [21]. In this study, 69.4% of patients showed mild disease, and 30.6% had moderate severity. It was observed that the patient's blood group did not have any effect on their presenting severity (p = 0.62); a similar observation was reported by Latz et al. [13]. Although a majority of the patients in this study recovered, five patients died. As the number of patients was low (1.3%), no analysis was performed regarding the death of the patients. Among them, one patient was of blood group A, two were of blood group B, and two were of blood group O. Four patients died in the hospital due to respiratory failure resulting from COVID pneumonia; one died at home, and the etiology was not determined. The median duration of the symptoms was 12 days. The blood group did not have any influence on the symptom duration. However, Zietz [15] described that the blood group A had less severity of illness, less intubation, and less mortality. More than 50% of the patients recovered within seven days. In the present study, 28% of the patients' symptoms persisted for more than 12 days. It was determined that the blood group did not have any influence on patient recovery. Approximately 16% of the patients converted to the next level of severity, that is, from mild-to-moderate, severe, and death. The blood group did not show any association with the conversion. Approximately 13.7% of the patients remained positive for COVID-19 in RT-PCR tests even after 14 days of initial positivity. The blood group A had a higher relative risk of delayed seroconversion. Latz et al [13], Zeng X et al [16] found no differences in their study. Approximately 44% developed different post-COVID symptoms. It was observed that post-COVID fatigue was the most prominent symptom. We did not find any significant

relationship between the blood groups and post-COVID syndrome. This study was performed on patients > 18 years of age with mild-to-moderate severity of illness. Therefore, our findings cannot be generalized. Therefore, this should be interpreted with caution.

This study has a few limitations. This was a single-center study, and the sample size was limited. We did not include the severe cases, and most of the follow-up was conducted through virtual interviews, which has some inherent drawbacks for the study.

## Conclusions

The prevalence of blood group A was higher among COVID-19 patients. The ABO blood group was not associated with the presentation and recovery period of COVID-19. However, the patients of blood group A had a higher risk of persistent positivity. To detect the susceptibility of the infection, further studies on the individuals with confirmed exposure to COVID-19 infection should be conducted.

## Supporting information

**S1 File. A protocol.**
(DOCX)

**S2 File. A certificate from Editage.**
(PDF)

**S3 File. Data sheet.**
(XLS)

**S4 File. Manuscript edited by Editage with track changes.**
(DOCX)

## Acknowledgments

I am grateful to every patient who gave their valuable consent for participation in this study; without their help, it would have been impossible to conduct this study.

We would like to thank Editage (www.editage.com) for English language editing.

## Author Contributions

**Conceptualization:** Reaz Mahmud, Farhana Binte Monayem, Mohammad Abdullah Yusuf, K. M. Nazmul Islam, Mohammad Zaid Hossain, Ahmed Hossain Chowdhury, A. K. M. Humayon Kabir, Kazi Gias Uddin Ahmed, Md. Mujibur Rahman.

**Data curation:** Reaz Mahmud, Mohammad Aftab Rassel, Farhana Binte Monayem, S. K. Jakaria Been Sayeed, Md Shahidul Islam, Mohammed Monirul Islam, Mohammad Abdullah Yusuf, Sabrina Rahman, Imran Mahmud, A. K. M. Humayon Kabir.

**Formal analysis:** Reaz Mahmud, Mohammad Aftab Rassel, Farhana Binte Monayem, S. K. Jakaria Been Sayeed, Md Shahidul Islam, Mohammed Monirul Islam, Mohammad Abdullah Yusuf, Sabrina Rahman, K. M. Nazmul Islam, Imran Mahmud, Md. Mujibur Rahman.

**Funding acquisition:** Reaz Mahmud, Kazi Gias Uddin Ahmed, Md. Mujibur Rahman.

**Investigation:** Reaz Mahmud, Mohammad Aftab Rassel, Farhana Binte Monayem, S. K. Jakaria Been Sayeed, Md Shahidul Islam, Mohammed Monirul Islam, Sabrina Rahman, K. M. Nazmul Islam, Imran Mahmud, Mohammad Zaid Hossain, Ahmed Hossain Chowdhury, A. K. M. Humayon Kabir, Md. Mujibur Rahman.

**Methodology:** Reaz Mahmud, Mohammad Aftab Rassel, Farhana Binte Monayem, S. K. Jakaria Been Sayeed, Md Shahidul Islam, Mohammed Monirul Islam, Mohammad Abdullah Yusuf, Sabrina Rahman, K. M. Nazmul Islam, Imran Mahmud, Mohammad Zaid Hossain, Ahmed Hossain Chowdhury, A. K. M. Humayon Kabir, Kazi Gias Uddin Ahmed, Md. Mujibur Rahman.

**Project administration:** Reaz Mahmud, Mohammad Aftab Rassel, S. K. Jakaria Been Sayeed, Ahmed Hossain Chowdhury, A. K. M. Humayon Kabir, Kazi Gias Uddin Ahmed, Md. Mujibur Rahman.

**Resources:** Reaz Mahmud, Farhana Binte Monayem, Md Shahidul Islam, Kazi Gias Uddin Ahmed, Md. Mujibur Rahman.

**Software:** Reaz Mahmud, Mohammad Abdullah Yusuf, K. M. Nazmul Islam.

**Supervision:** Reaz Mahmud, Ahmed Hossain Chowdhury, Kazi Gias Uddin Ahmed, Md. Mujibur Rahman.

**Validation:** Reaz Mahmud, S. K. Jakaria Been Sayeed, Md Shahidul Islam, Mohammed Monirul Islam, Mohammad Abdullah Yusuf, Imran Mahmud, Mohammad Zaid Hossain, Md. Mujibur Rahman.

**Visualization:** Reaz Mahmud, S. K. Jakaria Been Sayeed, Md Shahidul Islam, Mohammed Monirul Islam, Mohammad Abdullah Yusuf, Sabrina Rahman, Imran Mahmud, Mohammad Zaid Hossain, Md. Mujibur Rahman.

**Writing – original draft:** Reaz Mahmud, Mohammad Aftab Rassel, Farhana Binte Monayem, Mohammed Monirul Islam, Sabrina Rahman, Imran Mahmud.

**Writing – review & editing:** Reaz Mahmud, S. K. Jakaria Been Sayeed, Md Shahidul Islam, Mohammad Abdullah Yusuf, K. M. Nazmul Islam, Mohammad Zaid Hossain, Ahmed Hossain Chowdhury, A. K. M. Humayon Kabir, Kazi Gias Uddin Ahmed, Md. Mujibur Rahman.

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
