## [Decision Letter · Decision Letter 0]

4 Feb 2021

PONE-D-20-36898

Impact of ABO Blood Groups on Presentation and Outcomes of Confirmed SARS Cov-2 infection: A prospective study in a largest COVID-19 Dedicated Hospital in Bangladesh.

PLOS ONE

Dear Dr. Mahmud,

Thank you for submitting your manuscript to PLOS ONE. After careful consideration, we feel that it has merit but does not fully meet PLOS ONE’s publication criteria as it currently stands. Therefore, we invite you to submit a revised version of the manuscript that addresses the points raised during the review process.

This paper is potentially interesting, but it needs extensive revision before we can reconsider it again.

We look forward to receiving your revised manuscript.

Kind regards,

Prof. Raffaele Serra, M.D., Ph.D

Academic Editor

PLOS ONE

Journal Requirements:

3. We noted several instances of p = 0.00 in your manuscript. To comply with PLOS ONE submission guidelines, please report exact p-values for all values greater than or equal to 0.001. P-values less than 0.001 may be expressed as p < 0.001.

For more information on PLOS ONE's expectations for statistical reporting, please see https://journals.plos.org/plosone/s/submission-guidelines.#loc-statistical-reporting

4. Please describe in your methods section how capacity to provide consent was determined for the participants in this study.

Please also state whether your ethics committee or IRB approved this consent procedure.

If you did not assess capacity to consent please briefly outline why this was not necessary in this case.

5. Please amend either the title on the online submission form (via Edit Submission) or the title in the manuscript so that they are identical.

6. Please include captions for your Supporting Information files at the end of your manuscript, and update any in-text citations to match accordingly. Please see our Supporting Information guidelines for more information: http://journals.plos.org/plosone/s/supporting-information

Additional Editor Comments:

The manuscript is potentially interesting but several and substantial revisions are needed

Reviewers' comments:

Reviewer's Responses to Questions

**Comments to the Author**

1. Is the manuscript technically sound, and do the data support the conclusions?

Reviewer #1: Yes

Reviewer #2: Partly

2. Has the statistical analysis been performed appropriately and rigorously? 

Reviewer #1: Yes

Reviewer #2: No

3. Have the authors made all data underlying the findings in their manuscript fully available?

Reviewer #1: Yes

Reviewer #2: No

4. Is the manuscript presented in an intelligible fashion and written in standard English?

Reviewer #1: Yes

Reviewer #2: Yes

5. Review Comments to the Author

Reviewer #1: Good clinical report. Noticed patients were young! What were the causes of death of 5 patients? Do you have any long term follow up?

Reviewer #2: Here is a list of specific comments. Note: without line and page numbering, it was difficult to reference the reviews.

1. Because “impact” inferred causality, I suggest replacing “impact” with ‘association’ throughout the manuscript.

2. Abstract, Results, “the Presenting age . . . did not differ among the groups”: “The groups” were not clear in Abstract. Did it refer to the blood groups?

3. Materials and Methods, “this single centered ...outcome of Confirmed COVID-19 ...”: I suggest explicitly defining the outcome.

4. Materials and Methods, “for the estimation we grouped A . . . ”: I suggest including brief information about Groups I and II in Abstract.

5. Materials and Methods, “the estimated required sample size . . . ”: For the calculation of the required sample size, I suggest including the proportions of Groups I and II, the reference risk (i.e., risk in Group II), and the test statistic. Also, please specify the risk referred to the risk of recovery.

6. Statistical Analysis, “sample size 377 ...”: The sentence regarding the required sample size had been previously mentioned. I suggest excluding it.

7. Statistical Analysis, “to compare the groups A, B, AB and O . . . ”: The groups of A, B, AB and O contradicted the previously defined Groups I and II in the Materials and Methods section. I suggest using the same definitions of blood groups consistently throughout the manuscript.

8. Statistical Analysis, “the difference in median time-to-recovery . . . ”: For recovery, death was a competing event. Instead of stating “cox regression analysis”, I suggest stating ‘cause-specific hazards models’. No change to the models; it was just terminology. However, the use of Kaplan-Meier method and log-rank test were different. I suggest replacing them with ‘cumulative incidence functions method’ and ’Gray’s test’.

9. Results, “438 patients were enrolled . . . ”: Because survival analysis was used, the 52 lost-to-follow-up patients could be included in the analyses.

10. Results, “maximum follow up days was 45”: Weren’t the patients followed up to 30 days?

11. Table 1:

(11a) I suggest adding a column including Group II (i.e., blood groups B, AB and O) and a column of p-values comparing Groups I and II.

(11b) I suggest revising the column header “significance” as ‘p-values’.

12. Table 2: Please clarify in the table what the “Risk ratio/Hazard ratio” column referred to. Per the interpretation in the footnote, it seemed to refer the interactions between blood groups and attributes. If so, I suggest reporting hazard ratios for attributes in Group I and Group II because hazard ratios for interactions were not interpretable directly.

6. PLOS authors have the option to publish the peer review history of their article (what does this mean?). If published, this will include your full peer review and any attached files.

Reviewer #1: No

Reviewer #2: No

---

## [Author Response · Author response to Decision Letter 0]

10 Mar 2021

A rebuttal letter

To 

Editor in chief,

PLOS ONE

Subject: In response to review of the manuscript entitled “Association of ABO blood groups with presentation and outcomes of confirmed SARS CoV-2 infection: A prospective study in the largest COVID-19 dedicated hospital in Bangladesh”.

Dear Sir,

Thank you for reviewing my manuscript. I have tried to address each point raised by the academic editor and the reviewers.

Response to Academic editor

 PLOS ONE style-I have revised the manuscript according to PLOS ONE style.

 The manuscript was edited by Editage. A copy was uploaded as supporting information. A clean copy of edited manuscript was uploaded as manuscript file.

 P value =0.00 was mentioned in 2 occasion in the manuscript. I have corrected it to p <0.001 in the result section of the abstract and the discussion part of the manuscript.

Discussion- A one-sample t-test revealed a p value of < 0.001. (Page-11, line 205)

Abstract- The prevalence of blood group A [144 (32.9%)] was significantly higher among COVID-19 patients than in the general population (p < 0.001). (Page 2, line 36, 37)

 Description about How capacity to give consent was determined was added in the methodology section in the following way-

The capacity to provide consent was determined by the investigators in the presence of an attendant, a testimony of the attendant was obtained in the consent form, and the ethical review committee approved this consent procedure during the approval of the protocol. Those who were minors or unable to provide consent were excluded from the study. (Page-5, line 87-91)

 The title was amended as per reviewer’s advice both in the online submission form and manuscript as- “Association of ABO blood groups with presentation and outcomes of confirmed SARS CoV-2 infection: A prospective study in the largest COVID-19 dedicated hospital in Bangladesh”

 Caption of the supporting information file was added at the end of the manuscript.

Response to Reviewer’s:

 Is the manuscript technically sound, and do the data support the conclusions?

In response to Reviewer-2: I have tried to revise the manuscript according to your advice. In the subsequent section I shall explain it.

 Has the statistical analysis been performed appropriately and rigorously?

In response to Reviewer 2: Initially the statistical analysis was done with SPSS version 20. Some analysis you advised in the comment section like cumulative incidence function method was not possible in SPSS. So I revised the statistical analysis and did it with STATA version-15.1, StataCorp, 4905 Lakeway Drive, College Station, Texas 77845 USA. The change was made according to your advice which will be discussed in the subsequent sections. (Page-6, line 120-128)

 Have the authors made all data underlying the findings in their manuscript fully available?

In response to Reviewer 2: Data was made available to PLOS ONE and Dryad data repository, will be available to third party after the publication of the manuscript.

DOI https://doi.org/10.5061/dryad.dv41ns1xk

 Is the manuscript presented in an intelligible fashion and written in Standard English?

Both reviewers response was yes.

 Review comments to author:

 Reviewer 1: Thanks for the appraisal.

 Most of the patients were young: In this study the most of the study population were less than 40 years (about 60%).This is different from the Europe and USA. In USA older patients (ages ≥65 years) accounted for about 31% of all cases, in this study it was only 8%.This is probably due sociocultural background of Bangladesh. Here proportion of elderly population (5%) is less the western world (North America 16%, Europe 21%). More over many of our participants were health professionals.

 Cause of the death: total 4 of our patient died in Hospital due to respiratory failure due to COVID pneumonia. One of our patient died at home, the etiology was undetermined. ( Page-12, line 217-218)

 Long term follow up: No we don’t have long term follow up of all the patients.

 In response to Reviewer 2: Sorry for your discomfort. This was my second only writing to an international journal, I failed to understand the instruction. In the revised manuscript I have added the line and page numbering. Thank you for your suggestion, it helped to enrich my knowledge and views.

 Because “impact” inferred causality, I suggest replacing “impact” with ‘association’ throughout the manuscript.

Response: I have replaced the impact with association throughout the manuscript and renamed the title as Association of ABO blood groups with presentation and outcomes of confirmed SARS CoV-2 infection: A prospective study in the largest COVID-19 dedicated hospital in Bangladesh

 Abstract, Results, “the Presenting age . . . did not differ among the groups”: “The groups” were not clear in Abstract. Did it refer to the blood groups?

Response: I have clarified it in the revised manuscript and rewrite it as “The presenting age [mean (SD)] of group I [42.1 (14.5)] was higher than that of group II [38.8 (12.4), p=0.014]. Sex (p=0.23) and co-morbidity (hypertension, p=0.34; diabetes, p=0.13) did not differ between the patients in groups I and II”. ( Page 2, line 37-39)

Note: Previously we have compared A, B, AB, O and found no difference in the age of presentation. But this time when we compared between blood group A (group I) and other blood groups (group II) according to your suggestion, we found a significant difference. More over this time analysis was done with 438 patient including the lost to follow up patients.

 Materials and Methods, “this single centered ...outcome of Confirmed COVID-19 ...”: I suggest explicitly defining the outcome.

Response: In the revised manuscript the following lines are added to define the outcome explicitly- The outcomes in this research included: A. Duration required for clinical improvement, as defined below; B. Proportion of patients converted to the next level of severity; C. Proportion of the patients remaining positive for RT-PCR of COVID-19 on day 14 after the initial positivity; D. Development of post-COVID syndrome as defined below.(Page 4, line 79-82)

 Materials and Methods, “for the estimation we grouped A . . . ”: I suggest including brief information about Groups I and II in Abstract.

Response: According to your suggestion the following line was added in the abstract in the designs, settings part- We grouped participants with A-positive and A-negative blood groups into group I and participants with other blood groups into group II. (Page 2, line-33, 34)

 Materials and Methods, “the estimated required sample size . . . ”: For the calculation of the required sample size, I suggest including the proportions of Groups I and II, the reference risk (i.e., risk in Group II), and the test statistic. Also, please specify the risk referred to the risk of recovery.

Response: According to your suggestion we estimated our sample size as follows- An assumption that the expected proportions to be cured from COVID-19 by day 12 in group I (blood group A) and group II (blood groups B, O, AB) are 0.70 and 0.90, respectively. Thus, we required a total of 378 samples at a 1:2 ratio, which would provide a power of at least 90% in two-tailed tests and a p value less than 0.05, to detect significant differences between the groups. Therefore, considering a 10% dropout rate, we needed 416 samples in total. (Page 5, Line-93-96)

Note: formula used 

n=(r+1)/r (p^* (1-p^* ) 〖(Z_β+Z_(α/2))〗^2)/〖〖(p〗_1- p_2)〗^2 

r = ratio of group-1 and group - 2

p*= Average of proportion 

Zα/2 = Level of significance 

Zβ = Power of the test

p1 - p2= Effect size

p1 = Proportion in group - 1

p2 = Proportion in group - 2

Note: we were forced to assume the proportion as we found no suitable study to make a reference regarding the proportion of cure by day 12.

 Statistical Analysis, “sample size 377 ...”: The sentence regarding the required sample size had been previously mentioned. I suggest excluding it.

Response: It has been excluded

 Statistical Analysis, “to compare the groups A, B, AB and O . . . ”: The groups of A, B, AB and O contradicted the previously defined Groups I and II in the Materials and Methods section. I suggest using the same definitions of blood groups consistently throughout the manuscript.

Response: To compare the variables between group I (blood group A) and group II (other blood groups), an independent sample t-test was used. (page-6, line 121-122)

 Statistical Analysis, “the difference in median time-to-recovery . . . ”: For recovery, death was a competing event. Instead of stating “cox regression analysis”, I suggest stating ‘cause-specific hazards models’. No change to the models; it was just terminology. However, the use of Kaplan-Meier method and log-rank test were different. I suggest replacing them with ‘cumulative incidence functions method’ and ’Gray’s test’.

Response: I have re do the analysis according to your suggestion and rewritten the section as- The difference in the median time-to-recovery between group I and group II was determined with cumulative incidence by blood group from the Fine-Gray Model. The hazard ratios were calculated using cause-specific hazard models. Categorical variables are presented as n (%), normally distributed continuously as mean (SD), and skewed continuous variables as median (IQR). The statistical significance was set at P <0.05. (Page-6, line 124-128)

 Results, “438 patients were enrolled . . . ”: Because survival analysis was used, the 52 lost-to-follow-up patients could be included in the analyses.

Response: In the revised manuscript we analyze all 438 patients

 . Results, “maximum follow up days was 45”: Weren’t the patients followed up to 30 days?

Response: It was a mistake, omitted.

 Table 1:

(11a) I suggest adding a column including Group II (i.e., blood groups B, AB and O) and a column of p-values comparing Groups I and II.

(11b) I suggest revising the column header “significance” as ‘p-values’.

Response: Table I is reformed as-

Table 1: Baseline characteristics of 438 COVID19 patients with different Blood Group

12. Table 2: Please clarify in the table what the “Risk ratio/Hazard ratio” column referred to. Per the interpretation in the footnote, it seemed to refer the interactions between blood groups and attributes. If so, I suggest reporting hazard ratios for attributes in Group I and Group II because hazard ratios for interactions were not interpretable directly

Response: we split table 2 to table 2 and table 3 as follows. Table 3 variable were qualitative variable, so we could compute hazard ratio in this instances.

Table 2: Duration of illness of COVID-19 patients among Group-I and Group-II

Table 3: Outcome of COVID-19 patients among Group-I and Group-II

I hope I have tried my level best to address all of your point raised during the review process. Please consider my manuscript for publication in PLOS ONE.

Thanks

Dr. Reaz Mahmud

Assistant professor Neurology

Dhaka Medical College

---

## [Editor Report · Decision Letter 1]

15 Mar 2021

Association of ABO blood groups with presentation and outcomes of confirmed SARS CoV-2 infection: A prospective study in the largest COVID-19 dedicated hospital in Bangladesh

PONE-D-20-36898R1

Dear Dr. Mahmud,

We’re pleased to inform you that your manuscript has been judged scientifically suitable for publication and will be formally accepted for publication once it meets all outstanding technical requirements.

Kind regards,

Prof. Raffaele Serra, M.D., Ph.D

Academic Editor

PLOS ONE

Additional Editor Comments (optional):

amended manuscript is acceptable
---

## [Editor Report · Acceptance letter]

22 Mar 2021

PONE-D-20-36898R1 

Association of ABO blood groups with presentation and outcomes of confirmed SARS CoV-2 infection: A prospective study in the largest COVID-19 dedicated hospital in Bangladesh 

Dear Dr. Mahmud:

I'm pleased to inform you that your manuscript has been deemed suitable for publication in PLOS ONE. Congratulations! Your manuscript is now with our production department. 

Kind regards, 

on behalf of

Prof. Raffaele Serra 

Academic Editor

PLOS ONE